

# Blood parasites infecting the Hoatzin (*Opisthocomus hoazin*), a unique neotropical folivorous bird

M. Andreína Pacheco[1], M. Alexandra García-Amado[2], Jaime Manzano[3], Nubia E. Matta[3] and Ananias A. Escalante[1]

[1] Department of Biology/Institute for Genomics and Evolutionary Medicine (iGEM), Temple University, Philadelphia, PA, United States of America
[2] Laboratorio de Fisiología Gastrointestinal, Centro de Biofísica y Bioquímica, Instituto Venezolano de Investigaciones Científicas (IVIC), Caracas, Miranda, Venezuela
[3] Departamento de Biología, Grupo de Investigación Caracterización genética e Inmunología, Universidad Nacional de Colombia, Sede Bogotá-Facultad de Ciencias, Bogota, Colombia

Corresponding authors
M. Andreína Pacheco,
tug00270@temple.edu,
Maria.Pacheco@temple.edu
Ananias A. Escalante,
ananias.escalante@temple.edu

## ABSTRACT

The Hoatzin (*Opisthocomus hoazin*) is the only extant member of the order Opisthocomiformes. This unique South American bird lives in the riparian lowland vegetation characteristic of the Amazon and Orinoco basins. Hoatzins nest in communal social units close to water bodies; they are strictly folivores being the only bird with pregastric fermentation in the crop. Because of the complex logistics involved in capturing this bird, there is a knowledge gap on its parasites. This study documents two distant lineages of haemosporidian parasites (*Plasmodium* spp.) in a juvenile and two adults sampled in the Cojedes state, Venezuela. Although negative by microscopy, the parasite identification was possible by using molecular methods. We estimated the phylogenetic relationships on the parasite cytochrome b (*cytb,* 480 bp) gene and the mitochondrial DNA. We found one of the parasites lineages in two individuals (nestling and adult), and the corresponding fragment of *cytb* was identical to a one found in Wood Stork (*Mycteria americana*) from Brazil. The other lineage, found in an adult, has an identity of 469 out of 478 bp (98%) with *Plasmodium* sp. GAL-2012 (isolate THAMB08) from Brazil. Although a morphological description of these parasites was not possible, this is the first molecular study focusing on Hoatzin haemosporidian parasites and the first documentation of *Plasmodium* infections in the Hoatzin from Venezuela. Furthermore, we reported microfilaria in two adults as well as hematological parameters for six individuals. Information on hematological parameters could contribute to establishing the necessary baseline to detect underlying conditions, such as infections, in this bird species.

## INTRODUCTION

The Hoatzin (*Opisthocomus hoazin*), the only extant species of the Order Opisthocomiformes, is a unique bird native to the Orinoco and the Amazon basins in South America that includes Bolivia, Colombia, Ecuador, Peru, Venezuela, Brazil, and the lowlands of the

Guianas. Little is known about its ecology and evolution. The lack of information is due in part to difficulties accessing its habitat, as well as the behavior and size of these birds that makes them hard to capture. In Venezuela, this species is distributed along rivers of the central savannas to the eastern Orinoco River. They live in colonies in riverine and swamp forest, vegetation at edges of ponds, oxbow lakes, and other freshwater wetlands. It is folivore bird that feeds on riverine tree species, and nests in communal social units building their nest close to water bodies (*Thomas, 1996*). Its phylogenetic relationship remains an enigma (*Jarvis et al., 2014*; *Claramunt & Cracraft, 2015*; *Prum et al., 2015*). However, given that is the only bird with pregastric fermentation in the crop like that in ruminants (*Grajal et al., 1989*), most of the knowledge about this species is on the crop microorganism's community structure and ecology (e.g., *Godoy-Vitorino et al., 2010*; *Godoy-Vitorino et al., 2012*; *Bardele et al., 2017*). Indeed, there are only a few studies on ectoparasites (*Hernandes & Mironov, 2015*; *Bauchan et al., 2017*) and hemoparasite infections in the Hoatzin (*Renjifo, Sanmartin & De Zuleta, 1952*; *Gabaldon, 1998*). Only filarial parasite infections have been reported in the Hoatzin from Colombia (*Renjifo, Sanmartin & De Zuleta, 1952*), and no haemosporidian parasites had been found before in this species using blood films (*Renjifo, Sanmartin & De Zuleta, 1952*; *Gabaldon, 1998*).

In this study, we report two distant molecular lineages of haemosporidian parasites of the genus *Plasmodium* (Family Plasmodiidae, Order Haemosporida, Phylum Apicomplexa) found in the Hoatzin from the Cojedes River, a tributary of the Orinoco River in central Venezuela that is part of the Orinoquia. Plasmodiidae is a diverse group of vector-borne haemoparasites found in many terrestrial vertebrate hosts (*Garnham, 1966*; *Valkiūnas, 2005*; *Telford Jr, 2009*), including the species of human (*Cavalier-Smith, 2014*) and avian malaria (e.g., *Plasmodium relictum*; *Bensch, Hellgren & Pérez-Tris, 2009*; *Atkinson & Samuel, 2010*). This is the first evidence of *Plasmodium* infections in the Hoatzin in South America; additionally, we provided hematological parameters to generate information that may help to assess the health status of this bird species.

## MATERIAL AND METHODS

### Study area and samples

We caught eight individuals in *El Baúl* massif along the Cojedes River near the town of *El Baúl* located at the southwestern of the Cojedes state in the northcentral Venezuela. This is considered part of the Orinoquia region at the north of the Guyana Shield. *El Baúl* massif is about 720 km$^2$; it is relatively isolated and mountainous with steep topography following a northwest-southeast trend (*Viscarret, Wright & Urbani, 2009*). The vegetation is typical of the Orinoquia sedimentary and alluvial plains (e.g., vegetation of savannas, gallery forests, palm groves, and semi-deciduous forests). In the context of this study, the birds were captured in the gallery forest, which is moderately intervened and subject to seasonal flooding from the Cojedes River as well as forest fires (*González-Fernández et al., 2007*).

We trapped two individuals (one adult and one juvenile) in October 2010, and 6 individuals (two adult and four nestlings) in August 2015 in their nest during nighttime using butterfly nets and transported to the field laboratory. From each bird, we obtained

blood samples by brachial vein puncture, and then, immediately we prepared three to five thin smears. We preserved the rest of the sample in protein saver cards (Whatman 903, Whatman[TM], Cardiff, UK) for molecular analysis. We collected the specimens under permit number 0950 issued by the Venezuelan government (*Oficina Nacional de Diversidad Biológica, Ministerio del Poder Popular para Ecosocialismo, Hábitat y Vivienda*). All the animal protocols were approved by the ethics committees of *Instituto Venezolano de Investigaciones Científicas* (IVIC, Venezuela) under the number COBIANIM Dir-0885/1517/2014.

## Examination of blood films

Smears were air-dried immediately after preparation, fixed in absolute methanol for 5 min, and then stained with Giemsa (pH 7.2) for 45 min. Using a Leica DM750e microscope (Leica Microsystems, Heerbrugg, Switzerland), we first examined blood slides at $\times 400$ for 10 min and then at $\times 1,000$ for 20 min. For the capture of digital images, we scanned entirely those slides with hemoparasites using a Leica EC3 digital camera and processed with the LAS EZ (Leica Microsystems Suiza Limited, 2012). Then, we estimated the intensity of infection as No. of parasites/10,000 erythrocytes from erythrocyte counts with an increase of $\times 1,000$, focusing on areas where blood cells formed a monolayer (*Muñoz et al., 1999*). In addition, using ImageJ software (*Schneider, Rasband & Eliceiri, 2012*), we performed morphometric analyses of the erythrocytes. We measured the maximum cell width and length, as well as nuclear width and length for 30 erythrocytes per slide and per individual, following *Hartman & Lessler (1963)*. To estimate the percentage of each type of white blood cell present in blood, we also measured a differential white blood cell (WBC) count per 100 cells using the blood samples collected in 2015 following the protocol by *Clark, Boardman & Raidal (2009)*. We obtained all these measures for only those individuals caught in 2015 ($N = 6$) because of the better quality of their blood films.

## Molecular diagnostic of haemosporidian parasites

We extracted genomic DNA from whole blood using QIAamp® DNA Micro Kit (Qiagen GmbH, Hilden, Germany). We screened each sample for haemosporidian parasites by using a nested polymerase chain reaction (PCR) protocol that targets the parasite mitochondrial cytochrome b (*cytb*, 1,131 bp) gene using the primers described in *Pacheco et al. (2011)*; *Pacheco et al. (2018)*. The *cytb* external primers were forward AE298 5′-TGT AAT GCC TAG ACG TAT TCC 3′ and reverse AE299 5′-GT CAA WCA AAC ATG AAT ATA GAC 3′, and the internal primers were forward AE064 5′-T CTA TTA ATT TAG YWA AAG CAC 3′ and reverse AE066 5′-G CTT GGG AGC TGT AAT CAT AAT 3′. The primary PCR amplifications were carried out in 50 μl volume reaction using 5-8 μl of total genomic DNA, 2.5 mM MgCl$_2$, 1 × PCR buffer, 1.25 mM of each deoxynucleoside triphosphate, 0.4 mM of each primer, and 0.03 U/μl AmpliTaq polymerase (Applied Biosystems, Roche-USA). The primary PCR conditions were: A partial denaturation at 94 °C for 4 min and 36 cycles with 1 min at 94 °C, 1 min at 53 °C and 2 min extension at 72 °C, and we added a final extension of 10 min at 72 °C in the last cycle. Then, the nested PCRs were also made in 50 μl volume reaction using only 1 μl of the primary PCRs, 2.5 mM MgCl$_2$, 1× PCR buffer,

1.25 mM of each deoxynucleoside triphosphate, 0.4 mM of each primer, and 0.03 U/μl AmpliTaq polymerase. The nested PCR conditions were: A partial denaturation at 94 °C for 4 min and 25 cycles with 1 min at 94 °C, 1 min at 56 °C and 2 min extension at 72 °C, and we also added a final extension of 10 min at 72 °C in the last cycle. Both strands for all the *cytb* fragments were directly sequenced using an Applied Biosystems 3730 capillary sequencer. We identified all the *cytb* fragments obtained here as *Plasmodium* using BLAST (*Altschul et al., 1997*).

For those samples that were positive using the *cytb* PCR protocol, we amplified between 5,515 to 5,838 bp of the parasite mitochondrial genomes (mtDNA) using a nested PCR with Takara LA Taq™ Polymerase (TaKaRa Takara Mirus Bio) following manufacturers' directions. This fragment of the mtDNA included the three nonprotein coding regions between the ORFs (fragmented SSU rRNA and LSU rRNA) and the three protein-coding genes (*Cox3*, *Cox1* and *Cytb*) so only three tRNAs (7, 11, and 14) and two fragments of small subunit ribosomal RNAs (5 and 7) are missing. Oligos forward AE170 5′ GAGGATTCTCTCCACACTT CAATTCGTACTTC 3′ and reverse AE171 5′ CAGGAAAATWA TAGACCGAACCTTGGACTC 3′ were used for the primary PCR and internal oligos forward AE176 5′ TTTCATCCTTAAATCTCGTAAC 3′ and AE136 reverse 5′ GACCGAA CCTTGGACTCTT 3′ for the inner PCR. The PCR conditions were a partial denaturation at 94 °C for 1 min and 30 cycles with 30 s at 94 °C and 7 min at 68 °C and a final extension of 10 min at 72 °C. Then, we excised two independent PCR products (50 ul) from the gel (bands of approximately 6 kbp) and purified using QIAquick® Gel extraction kit (Qiagen, GmbH, Hilden, Germany). We cloned at least two independent PCR products using pGEM®-T Easy Vector Systems (Promega, Madison, WI, USA), and we sequenced four clones from each individual. We sequenced both strands for PCR products and clones using an Applied Biosystems 3730 capillary sequencer. Given that the *cytb* partial sequences and the *cytb* gene from the mtDNA genome were 100% identical for each sample, we only deposited the mtDNA genome sequences in GenBank under the accession numbers KY653749 to KY653751.

## Phylogenetic analysis of the *cytb* fragment and mtDNA genome

We performed two different nucleotide alignments by using ClustalX v2.0.12 and Muscle as implemented in SeaView v4.3.5 (*Gouy, Guindon & Gascuel, 2010*) with manual editing. The first alignment was constructed with 45 *cytb* partial sequences (480 bp) belonging to three genera (*Leucocytozoon*, *Haemoproteus*, and *Plasmodium*). This alignment included the sequences obtained in this study as well as sequences from well-known parasite species based on morphology (*Valkiūnas & Iezhova, 2018*) that were available in GenBank (*Benson et al., 2012*) and MalAvi (*Bensch, Hellgren & Pérez-Tris, 2009*) databases at the time of this study. Sequences that showed a similarity >95% using BLAST (*Altschul et al., 1997*) were also included even when they are not clearly linked to the described species. The second alignment (5,286 bp excluding gaps) was done using 31 mtDNA genome sequences belonging also to the three genera, including the sequences obtained in this study and sequences from well-known parasite species (using morphology) available in GenBank. Subsequently, the alignment was divided into six partitions corresponding to the three

non-protein coding regions between the ORFs (fragmented SSU rRNA and LSU rRNA) and the three protein-coding genes, keeping their order in the mtDNA genome (nonprotein coding, *cox3*, nonprotein coding, *cox1*, *cytb*, nonprotein coding).

Then, we inferred phylogenetic hypotheses based on the first (partial *cytb* gene) and second (mtDNA genome) alignments using the Bayesian methods implemented in MrBayes v3.2.6 with the default priors (*Ronquist & Huelsenbeck, 2003*). To estimate the phylogenetic hypothesis that best fit the data, we used the general time reversible model with gamma-distributed substitution rates and a proportion of invariant sites (GTR $+\Gamma+$ I) on the *cytb* alignment and for each partition in the mtDNA genome alignment. This model was the one with the lowest Bayesian Information Criterion (BIC) scores for both alignments and each partition as estimated by MEGA v7.0.14 (*Kumar, Stecher & Tamura, 2016*). We inferred Bayesian support for the nodes in MrBayes by sampling every 1,000 generations from two independent chains lasting $4 \times 10^6$ Markov Chain Monte Carlo (MCMC) steps. The chains were assumed to have converged once the value of the potential scale reduction factor (PSRF) was between 1.00 and 1.02 and the average SD of the posterior probability was < 0.01 (*Ronquist & Huelsenbeck, 2003*). Then, we discarded 25% of the sample once convergence was reached as a "burn-in". For both phylogenies, we used *Leucocytozoon* species as out-group. Genbank accession numbers for all sequences used in the analyses are given in the phylogenetic trees.

## RESULTS

In this study, one sample was positive for haemosporidian parasites by microscopy (adult caught in 2015). The individual had only *Plasmodium* sp. trophozoites in its blood films (Fig. 1), so a morphological description of this parasite was not possible. Indeed, the parasitemia was very low (<0.01), consistent with a subpatent infection. In addition, the two adults caught in 2015 were infected with microfilariae, so one of the adults has a coinfection of filarial parasites and *Plasmodium* sp. (Fig. 1). Unfortunately, the identification of nematode microfilariae at species level is difficult given their high degree of morphological and morphometric similarities (*McKeand, 1998*). None of the nestlings were infected with hemoparasites. Morphometry of the uninfected erythrocytes is shown in Table 1. We compared these measurements with those from the erythrocytes of other avian species with similar body size. In addition, we provided differential white blood cell (WBC) count profiles in Table 2.

The molecular diagnostic detected that the two individuals caught in 2010 (one adult and one juvenile) and one of the adults caught in 2015 (the one positive by microscopy) were positive by nested PCR (3/8, 37.5%). In order to characterize these *Plasmodium* species, for those individuals ($N = 3$) that were positive by nested PCR, we obtained the parasite mtDNA genome sequences. We further examined these sequences by using phylogenetic analyses (Fig. 2) yielding two estimated gene trees, one just with partial *cytb* sequences commonly used to identify haemosporidia parasites (*Bensch, Hellgren & Pérez-Tris, 2009*; *Pacheco et al., 2018*) and the other with mtDNA genome (*Pacheco et al., 2018*). These phylogenies have similar topologies. We found two lineages of *Plasmodium* (identified as 1

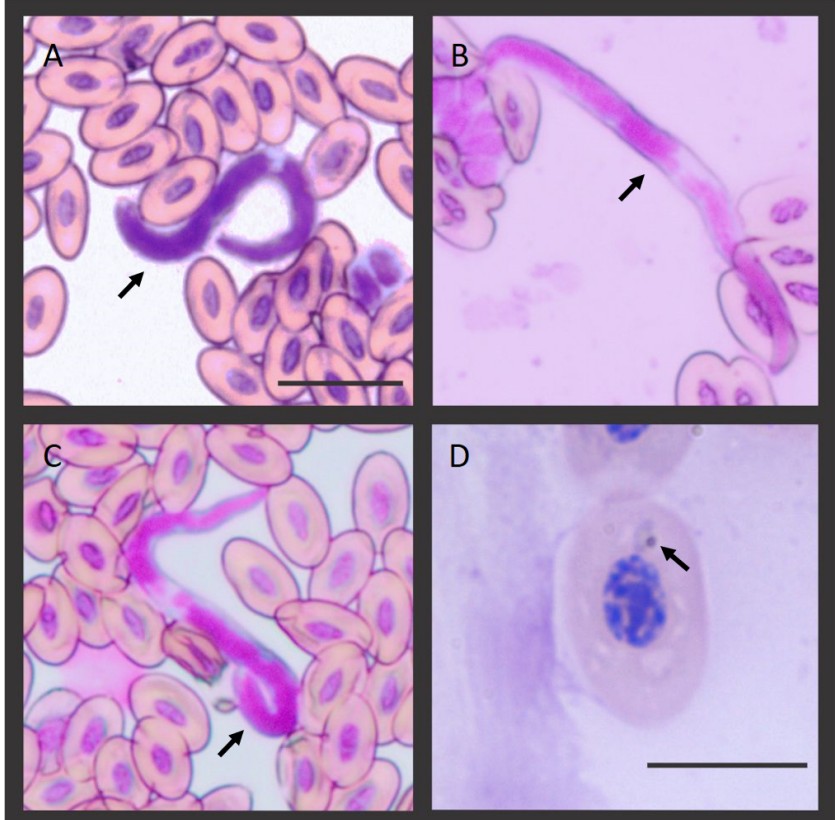

**Figure 1** **Microfilaria and *Plasmodium* sp. blood stage infecting the Hoatzins (*O. hoazin*) caught in August 2015 at the Cojedes River, Venezuela.** (A) Hoatzin individual infected only with filarial parasites. (B\*), (C\*), and (D&) Hoatzin individual infected with filarial parasites and *Plasmodium* sp. \*Microfilarial stages of the filarial parasite (Scale bar = 20 μm) and &*Plasmodium* sp. trophozoite (Scale bar = 10 μm). We indicated the hemoparasite stages by the black arrows.

and 2 in Fig. 2) in three individuals of Hoatzin. The individuals caught in 2010 (one adult and one juvenile) were infected with the same *Plasmodium* lineage (KY653750–KY653751), and their mtDNA genome sequences were 100% identical.

## DISCUSSION

This is the first report of *Plasmodium* species in this bird and the first record of microfilarias in Hoatzins from Venezuela. We have no sign of pathogenesis associated with these infections. However, as numbers of heterophils and lymphocytes could be affected by stress such as a parasitic infection, the ratio of one to the other (H:L) is commonly used as a stress indicator (*Gross & Siegel, 1983*; *Maxwell, 1993*). Thus, we reported the H:L ratio since it is expected to increase in response to stressors such as infectious diseases (*Davis, Maney & Maerz, 2008*), especially in birds with high parasitemia (*Granthon & Williams, 2017*). Here, the bird infected with filarial parasites and *Plasmodium* sp. had an increase in the number of lymphocytes and a decrease in the number of heterophils in comparation with the haemosporidian negative birds (infected only with microfilaria and uninfected

**Table 1 Comparison of erythrocyte measurements from different bird orders including the values for Hoatzin.** The values for Hoatzin (O. hoazin) are from six individuals caught in August 2015 at the Cojedes River, Venezuela.

| | Cytosome | | | Nucleus | | |
| --- | --- | --- | --- | --- | --- | --- |
| | Length (μm) | Width (μm) | Ratio (L/W) | Length (μm) | Width (μm) | Ratio (L/W) |
| Opisthocomiformes | | | | | | |
| *O. hoazin* | 15.3 ± 0.85 | 8.7 ± 0.71 | 1.76 | 6.2 ± 0.61 | 3.4 ± 0.47 | 1.82 |
| Gruiformes[a] | | | | | | |
| *Rallus elegans* | 14.5 ± 0.32 | 7.7 ± 0.17 | 1.89 | 5.7 ± 0.13 | 2.9 ± 0.12 | 1.97 |
| *Aramides cajanea* | 12.9 ± 0.78 | 7.2 ± 0.50 | 1.79 | 5.5 ± 0.40 | 3.3 ± 0.37 | 1.67 |
| *Fulica americana* | 11.4 ± 0.26 | 7.5 ± 0.22 | 1.51 | 4.2 ± 0.06 | 2.3 ± 0.07 | 1.83 |
| Charadriiformes[a] | | | | | | |
| *Jacana spinosa* | 13.7 ± 0.79 | 7.5 ± 0.52 | 1.83 | 5.9 ± 0.64 | 3.7 ± 0.33 | 1.59 |
| *Charadrius wilsonia* | 12.8 ± 0.27 | 7.3 ± 0.13 | 1.76 | 5.8 ± 0.11 | 2.4 ± 0.05 | 2.41 |
| *Himantopus mexicanus* | 12.8 ± 0.16 | 6.9 ± 0.18 | 1.85 | 5.8 ± 0.20 | 2.5 ± 0.99 | 2.32 |
| Accipitriformes[a] | | | | | | |
| *Accipiter cooperii* | 14.3 ± 0.27 | 8.1 ± 0.13 | 1.77 | 6.2 ± 0.14 | 2.4 ± 0.11 | 2.58 |
| *Buteo platypterus* | 13.4 ± 0.51 | 7.6 ± 0.34 | 1.77 | 6.2 ± 0.31 | 3.0 ± 0.32 | 2.07 |
| *Rupornis magnirostris*[b] | 13.1 ± 0.72 | 7.4 ± 0.36 | 1.78 | 6.83 ± 0.38 | 2.84 ± 0.18 | 2.4 |

**Notes.**

[a] Erythrocyte measurements are from *Hartman & Lessler (1963)*.

[b] from *Tostes et al. (2017)*.

**Table 2 Differential white blood cell (WBC) counts and H:L ratio in the Hoatzins ($N = 6$) caught in August 2015 at the Cojedes River, Venezuela.**

| | Uninfected nestlings ($N = 4$) | Infected adults microfilaria ($N = 1$) | microfilaria/*Plasmodium* ($N = 1$) |
| --- | --- | --- | --- |
| Heterophils | 64.25 ± 2.75 | 67 | 53 |
| Lymphocytes | 27 ± 1.83 | 17 | 35 |
| Monocytes | 5 ± 0.82 | 8 | 5 |
| Eosinophils | 3.25 ± 0.5 | 7 | 5 |
| Basophils | 0.5 ± 0.58 | 1 | 1 |
| H:L | 2.4 | 3.9 | 1.5 |

individuals), and so a lower H:L ratio (1.5 vs. 2.4, Table 2). The opposite occurred in the individual infected only with filarial parasites (H:L = 3.9 vs. 2.4, Table 2). WBC counts are difficult to interpret within and between species (e.g., *Ricklefs & Sheldon, 2007*) so these results should be taken with caution especially considering that these are a few individuals from a small sample. Considering the paucity of data from this species, we provided the WBC count measures for comparison in future studies.

It has been hypothesized an association between erythrocyte size and the species body size as result of differences in their metabolic rates (*Hartman & Lessler, 1963*). As expected, we found that erythrocyte size from Hoatzins was comparable to those reported from putative sister taxa like Gruiformes (*Jarvis et al., 2014*; *Claramunt & Cracraft, 2015*), Charadriiformes (*Claramunt & Cracraft, 2015*), and Accipitriformes (*Prum et al., 2015*)

Peer J

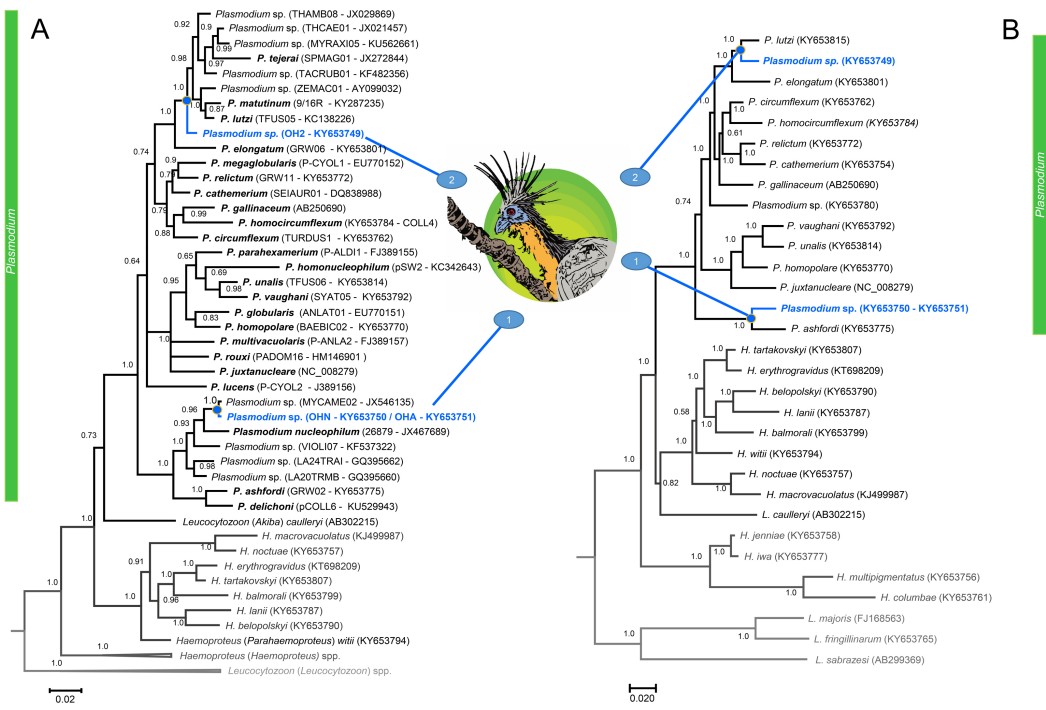

**Figure 2** **A Bayesian phylogenetic hypothesis of *Plasmodium* parasites infecting the Hoatzins (*O. hoazin*) caught at the Cojedes River, Venezuela.** We constructed phylogenetic trees based on parasites (A) partial sequences of the *cytb* gene (45 sequences and 480 bp excluding gaps) and mtDNA genomes (31 sequences and 5286 bp excluding gaps). The values above branches are posterior probabilities (see "Material and methods"). *Leucocytozoon* genus (outgroup) is indicated in grey. We provided in parentheses both lineages (as deposited in the MalAvi database) and their Genbank accession numbers for all the sequences used in the analyses.

(Table 1). Furthermore, differential white blood cell (WBC) count profiles (Table 2) from non-infected individuals showed marked similitude with those reported in nonpasserine birds such as cranes, raptors, and vultures (*Davis, 2009*).

Wherever there are subpatent infections, it is difficult to determine the prevalence of haemosporidian parasites since those infections are usually submicroscopic. In such cases, parasites can only be detected by polymerase chain reaction using genes with high copy number like *cytb* (*Pacheco et al., 2018*). The molecular diagnostic detected two lineages of *Plasmodium* genus (Fig. 2) in these three individuals of Hoatzin. Interesting, the lineage found in the individuals caught in 2010 (one adult and one juvenile) has been only reported so far in Brazil (*Villar et al., 2013*; *Ferreira Jr et al., 2017*; *Tostes et al., 2017*) suggesting that it is distributed solely in South America. In particular, its partial *cytb* sequence match 100% with a parasite lineage reported in several bird species: MYCAME02 isolated from Wood Storks nestlings found in the northern region of Brazil (*Mycteria Americana*, Pelecaniformes)(*Villar et al., 2013*), H2 isolate from Streaked Flycatcher found in Southeastern Brazil (*Myiodynastes maculatus*, Passeriformes) (*Ferreira Jr et al., 2017*), and lineages reported from birds belonging to the orders Strigiformes, Accipitriformes,

and Falconiformes kept in captivity in Southeaster Brazil (*Tostes et al., 2017*). In the phylogenetic analysis, this lineage appears as the sister taxon of *Plasmodium nucleophilum*, parasite isolated from an Egyptian Goose in São Paulo Zoo, Brazil (*Chagas et al., 2013*). Both parasites, the lineage reported here and *P. nucleophilum*, are part of a monophyletic group that includes *Plasmodium ashfordi* and *Plasmodium delichoni*. This monophyletic group is at the base of the mitochondrial phylogeny of the known Avian *Plasmodium* parasites. Given the lack of gametocyte data, it is possible that the *Plasmodium* infections in these two Hoatzin individuals were abortive, but the fact that the two individuals (caught together) were infected with the same parasite suggests that its transmission is occurring, and the parasite lineage is circulating in Cojedes State, Venezuela. *Tostes et al. (2017)* re-described the parasite linked to this *cytb* lineage as *Plasmodium* (*Novyella*) *paranucloephilum*, a species originally described by *Manwell & Sessler (1971)* in a South American tanager of uncertain species likely from northern Brazil. However, given the absence of morphological data in this study, and that there is no *cytb* neither other mtDNA genome sequences belonging to the original parasite description, we consider that it is premature to identify the lineage found in this study as *P. paranucloephilum*. The identity of this lineage could be established when more samples from birds with high parasitemia of this haemosporidian parasite become available.

Regarding the second lineage found infecting hoatzin (KY653749, Fig. 2), it appears as the common ancestor of a clade that includes *Plasmodium tejerai*, *Plasmodium matutinum* and *Plasmodium lutzi*. This clade forms a monophyletic group with *P. elongatum* (Fig. 2); one of the most pathogenic and generalist avian malaria parasite worldwide (*Palinauskas et al., 2016*). Given that we did not find any sequence with 100% similarity with our lineage in the available databases, this result indicates that likely a new parasite is circulating in the area. Considering the difficulty of catching this bird species, it is worth noticing that even with this small sample size (only eight individuals including four nestlings) we found two *Plasmodium* lineages infecting the Hoatzins.

Riparian zones from the Orinoco and Amazon basins are considered important in terms of their biodiversity since they result from variable flood regimes, geographically unique channel processes, altitudinal climate shifts, and upland influences on fluvial corridors. Furthermore, these areas are treated as critical habitat to several endangered bird species, refuges to the fauna inhabiting adjacent areas and, in some cases, hotspots and corridors for bird migration and dispersal (*Naiman & Decamps, 1997*; *Franchin et al., 2009*). In the Amazonia, the ability of avian malaria parasites to disperse geographically and shift among avian hosts have been played a role in their radiation and have shaped their current distributions and diversity (*Sebaio et al., 2012*; *Fecchio et al., 2018a*; *Fecchio et al., 2018b*). Thus, the fact that one of the lineages reported in the Hoatzin has been found also in species in Brazil is consistent with this notion of broad geographic distribution and multiple hosts. This finding also indicate that the parasite communities in the Orinoquia and the Amazon basin share species so, at this point, we can only speculate that similar processes may shape the parasite communities in both areas.

Riparian ecosystems in the Orinoquia are in a state of dynamic flux due to human interventions, seasonal flooding, and fires. In addition, these areas are suitable for insects

that could act as vectors. Considering all these factors, the Orinoquia should be given special priority for future research in order to document its parasite-avian host ecology and biodiversity.

## ACKNOWLEDGEMENTS

The authors express their sincere gratitude to the local personnel at the study sites that helped with trapping the birds. We thank Ariana Cristina Pacheco for the silhouettes design and the DNA laboratory at the School of Life Sciences (Arizona State University) for their technical support. We thank the Editor and the reviewers for their valuable comments.

### Funding

This research was supported in part by Temple University. The funders had no role in study design, data collection and analysis, decision to publish, or preparation of the manuscript.

### Grant Disclosures

The following grant information was disclosed by the authors:
Temple University.

### Competing Interests

The authors declare there are no competing interests.

### Author Contributions

- M. Andreína Pacheco conceived and designed the experiments, performed the experiments, analyzed the data, contributed reagents/materials/analysis tools, prepared figures and/or tables, authored or reviewed drafts of the paper, approved the final draft.
- M. Alexandra García-Amado performed the experiments, contributed reagents/materials/analysis tools, approved the final draft.
- Jaime Manzano performed the experiments, contributed reagents/materials/analysis tools, prepared figures and/or tables, approved the final draft.
- Nubia E. Matta contributed reagents/materials/analysis tools, approved the final draft.
- Ananias A. Escalante conceived and designed the experiments, analyzed the data, contributed reagents/materials/analysis tools, authored or reviewed drafts of the paper, approved the final draft.

### Animal Ethics

The following information was supplied relating to ethical approvals (i.e., approving body and any reference numbers):

All the animal protocols were approved by the ethics committees of Instituto Venezolano de Investigaciones Científicas (IVIC, Venezuela) under the number COBIANIM No. Dir-0885/1517/2014.
## Field Study Permissions

The following information was supplied relating to field study approvals (i.e., approving body and any reference numbers):

We collected the specimens under permit number 0950 issued by the Venezuelan government ("Oficina Nacional de Diversidad Biológica, Ministerio del Poder Popular para Ecosocialismo, Hábitat y Vivienda").

## Data Availability

The mtDNA genome sequences described here are accessible via GenBank accession numbers KY653749 to KY653751.

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
