# Peer review of "Blood parasites infecting the Hoatzin (Opisthocomus hoazin), a unique neotropical folivorous bird"

_PeerJ, doi:10.7717/peerj.6361_

## Round 0.1 · original submission · Major Revisions

Dear authors

Your ms has been reviewed, please respond to the comments. A major concern is the small sample size. Maybe, you could try to get more samples. ((By the way, I have over 20 blood samples in my sample collection)).

Kind regards

Michael Wink
Academic editor

Reviewer 1 ·

Basic reporting

The article offers the first information of haemosporidian parasites in the Hoatzin. Unfortunately the sample sizes are really small. The authors explain this by the rarity and difficulty to catch this species. There are a few other short-comings: The title of the paper is misleading as in the paper next to avian malaria other blood parasites are reported (but poorly identified, it is unclear what species). I suggest to change in the title Plasmodium to "Blood parasites" or, alternatively, leave out microfilaria and red blood cell data as this is limited information with regard to haemosporidian infection. If this would be my paper, I would concentrate on haemosporidian parasites. Maybe the blood parameter data is more suitable for another journal.
There are quite a few mistakes regarding the English language. This needs to be fixed.
Regarding references and background all is well. The article structure is also professional, only I would consider leaving out all tables and figures that do not relate to haemosporidian parasites. The haemosporidian tree is nice, but if at all possible it would be interesting to indicate hosts (at least for specialists).
So the article is not entirely self-contained and needs work (see above).

Experimental design

These points are all OK BUT a larger sample would really be important. Molecular work seems OK and also the analyses. See attached pdf for further comments.

Validity of the findings

See "basic reporting"

Additional comments

I hope you will be able to catch many more birds in Venezuela and check them for avian malaria. Your work is very good and relevant.

Annotated reviews are not available for download in order to protect the identity of reviewers who chose to remain anonymous.

Reviewer 2 ·

Basic reporting

Dear Editor,

This is a well-organized paper which provides a detailed analysis of parasites infecting Hoatzin birds, as a model organism. The Hoatzin is unique as a strictly folivorous bird species, although studies for this species are limited in various aspects, especially haemoparasites. The introduction and discussion are written with great clarity and objectivity. Considering the difficulties of catching these birds and the lack of studies for the floodplains of the Amazon, the sampling was reasonable to reach the aims of the study. The data analysis was well elaborated. The authors used different parasite detection and diagnostic methods such as blood film examination and molecular analysis. Discussion could be improved if the authors compare the level of parasites found in the floodplains to studies conducted in other biomes, such as Cerrado and Atlantic Forests, as found in publications by Miguel Ângelo Marini and Fabiane Sebaio. Other aspects that should be emphasized by the authors are the specific environmental vulnerabilities of the floodplains and how this could affect parasitism levels of birds as well as other organisms. The floodplains not only represent the second largest environment of the Amazon region (in territorial extension), but also shelter almost one-third of the bird species that inhabit the region. The avifauna of the floodplain presents a greater complexity. Several species are specialized in micro habitats of the floodplains (Remsen & Parker, 1983; Rossen, 1990) and it has been shown that avifauna of the floodplains of Amazonia are not homogeneously distributed (Cohn-Haft et al., 2007). Cohn-Haft et al., 2007 also show that, while birds in Varzea are specialized in microhabitats, they still present in environments modified by human habitation, suggesting that these birds are relatively tolerant or have not yet had sufficient time to feel the impacts of human actions. How parasitism could affect the birds is a field that is not currently well explored. But I suggest that the authors add a paragraph discussing the conservation implications of their findings regarding the levels of bird parasites found in their study as compared to other studies conducted in other biomes. The authors should also provide more detailed information on the degree of conservation of the areas where the birds were sampled and how the level of parasitism could increase if the areas are affected by anthropic disturbances (citing other studies on these subject). With the inclusion of the above modifications, I recommend the publication of this article in the journal PeerJ.

CONH-HAFT, M. ; NAKA, L. N. ; FERNANDES, A. M. 2007 . Padrões de distribuição da avifauna da várzea do rio Solimões-Amazonas. Bases científica para a conservação da várzea: identificação e caracterização de regiões. 1ed.: Ibama/ProVárzea, v. , p. 287-323.

Experimental design

Considering the difficulties of catching these birds and the lack of studies for the floodplains of the Amazon, the sampling was reasonable to reach the aims of the study. The data analysis was well elaborated. The authors used different parasite detection and diagnostic methods such as blood film examination and molecular analysis.

Validity of the findings

No comment

Additional comments

Discussion could be improved if the authors compare the level of parasites found in the floodplains to studies conducted in other biomes, such as Cerrado and Atlantic Forests, as found in publications by Miguel Ângelo Marini and Fabiane Sebaio. Other aspects that should be emphasized by the authors are the specific environmental vulnerabilities of the floodplains and how this could affect parasitism levels of birds as well as other organisms. The floodplains not only represent the second largest environment of the Amazon region (in territorial extension), but also shelter almost one-third of the bird species that inhabit the region. The avifauna of the floodplain presents a greater complexity. Several species are specialized in micro habitats of the floodplains (Remsen & Parker, 1983; Rossen, 1990) and it has been shown that avifauna of the floodplains of Amazonia are not homogeneously distributed (Cohn-Haft et al., 2007). Cohn-Haft et al., 2007 also show that, while birds in Varzea are specialized in microhabitats, they still present in environments modified by human habitation, suggesting that these birds are relatively tolerant or have not yet had sufficient time to feel the impacts of human actions. How parasitism could affect the birds is a field that is not currently well explored. But I suggest that the authors add a paragraph discussing the conservation implications of their findings regarding the levels of bird parasites found in their study as compared to other studies conducted in other biomes. The authors should also provide more detailed information on the degree of conservation of the areas where the birds were sampled and how the level of parasitism could increase if the areas are affected by anthropic disturbances (citing other studies on these subject). With the inclusion of the above modifications, I recommend the publication of this article in the journal PeerJ.

CONH-HAFT, M. ; NAKA, L. N. ; FERNANDES, A. M. 2007 . Padrões de distribuição da avifauna da várzea do rio Solimões-Amazonas. Bases científica para a conservação da várzea: identificação e caracterização de regiões. 1ed.: Ibama/ProVárzea, v. , p. 287-323.

·

Basic reporting

This manuscript reports on haemoparasite infections, apparently for the first time, on an Amazonian bird difficult to sample. The methods are consistent with current literature, but sample size is rather small, making this study very preliminary. The study also reports data on microfilaria and information on haematological parameters, both partially complimentary to haemoparasite infection, but mixed in the text. The manuscript could be improved by either (a) structuring it with three components (haemoparasite, microfilaria and haematological parameters) or (b) focusing even more on the haemoparasite infection and only appending at the end of the Results and Discussion sections the data on microfilaria and haematological parameters. Microfilaria is not mentioned in the title, but is cited first in the objective (line 63) and in the first paragraph of the Discussion (line 192). So, the manuscript could be re-structured, from the title, to incorporate microfilaria and haematological parameters also as its main objective, especially since the bird seems to be so difficult to capture and study.

Experimental design

No comment. See below.

Validity of the findings

No comment. See below.

Additional comments

Specific comments:
Lines 75-76. Describe shortly the vegetation characteristics and the conservation status of the sampling site since disturbance can affect parasitism. From a satellite image the study site seems very disturbed.
Lines 192-196. The Discussion starts with filarial infections, though it is secondary in the manuscript to haemosporidian infection. It should focus first on the most important main results.
Lines 204-205. This sentence needs references to support that “the ratio of one to the other (H:L) is commonly used as a stress indicator”. For example:
Gross and Siegel. 1983. Evaluation of the heterophil/lymphocyte ratio as a measure of stress in chickens. AVIAN DISEASES 27:972-979.
Maxwell 1993. Avian blood leukocyte responses to stress. WORLD’S POULTRY SCIENCE JOURNAL 49:34-43.

Line 221. Provide a reference for “has been only reported so far in Brazil”.
Lines 253-255. The statement “Overall, our results suggest that the parasite biodiversity in this species, and likely other birds inhabiting riparian ecosystems (e.g., Wood Stork and other species of Ciconiidae family), could be high” seems premature, especially considering the very small sample size. I recommend the following references to improve the discussion about malaria in Amazonian birds:

Belo et al. 2011. Host species and parasite lineage diversity of haemosporidians in three different environments with distinct levels of disturbance. Plos ONE 6: 17654.
Fecchio et al. 2017. Avian malaria, ecological host traits and mosquito abundance in southeastern Amazonia. PARASITOLOGY 44:1-16.
Fecchio et al. 2018. Host community similarity and geography shape the diversity and distribution of haemosporidian parasites in Amazonian birds. ECOGRAPHY 41:505-515.
Fecchio et al. 2018. Diversification by host switching and dispersal shaped the diversity and distribution of avian malaria parasites in Amazonia. OIKOS 127:1233-1242.
Folegatti et al. 2017. A systematic review on malaria sero-epidemiology studies in the Brazilian Amazon: insights into immunological markers for exposure and protection. MALARIA JOURNAL 16:107.

Other minor corrections:
Line 23. Replace Opisthocomiforme by Opisthocomiformes.
Line 97. Remove the comma (,) after Lessler.
Line 100. 2009 should be (2009).
Line 176. Replace hemoparasites by haemoparasites. Correct also in the abstract.
Line 202. Replace “no” with “non-” at the end of the line.

---

## Round 0.2 · accepted · Accept

Dear authors,

Good news! Your revision is adequate and we can publish your paper although sample size is quite low.

Congratulations,

Michael Wink
Academc Editor

# ·

Basic reporting

The manuscript which is not well known species the Hoatzin (Opisthocomus hoazin), is well designed and analyzed. The authors give us first results of blood parasites of the Hoatzin birds. Detail molecular analysis was done for identification the blood parasite of the birds. Although the manuscript improved according to previous reviewers’ suggestion and comment, still the main problem of the manuscript is sample size which is already mentioned previous reviewers.
Differential white blood cell counts are given Table 2 it is very interesting approach but microfilaria and microfilaria/Plasmodium results are given only one specimen. So, it is not useful any critics
Line 216 -220 it is impossible to make any critics like decrease or increase using one specimen information. The sentence should be rewrite

Experimental design

Catching the Hoatzin birds seems to be difficult but more samples could have found from the other lab. Lab works are ok

Validity of the findings

No comment

Reviewer 5 ·

Basic reporting

Interesting findings and complex discussion, though I am not convinced about the value of microfilariae reporting. The statement, that other studies also do so is not real justification. On the other hand, if this few sentences remain it causes no harm.

Experimental design

Part on Plasmodium parasites uses robust methodology, the basic haematology and microfilariae reporting are by-products of limited value. To be honest, presenting the haematological parameters with some clinical relevance would require larger set of individuals, more defined conditions, age, sex etc., as the haematological parameters vary a lot.

Validity of the findings

see above